# Portal Thrombosis in Cirrhosis: Role of Thrombophilic Disorders

**DOI:** 10.3390/jcm9092822

**Published:** 2020-08-31

**Authors:** José Ignacio Fortea, Inés García Carrera, Ángela Puente, Antonio Cuadrado, Patricia Huelin, Carmen Álvarez Tato, Paloma Álvarez Fernández, María del Rocío Pérez Montes, Javier Nuñez Céspedes, Ana Batlle López, Francisco José González Sanchez, Marcos López Hoyos, Javier Crespo, Emilio Fábrega

**Affiliations:** 1Gastroenterology and Hepatology Department, University Hospital Marqués de Valdecilla, 39008 Santander, Spain; ines.garciac@scsalud.es (I.G.C.); angelam.puente@scsalud.es (Á.P.); antonio.cuadrado@scsalud.es (A.C.); patricia.huelin@scsalud.es (P.H.); carmenalvtato@hotmail.com (C.Á.T.); palomaalvrz@gmail.com (P.Á.F.); javiercrespo1991@gmail.com (J.C.); emilio.fabrega@scsalud.es (E.F.); 2Group of Clinical and Translational Research in Digestive Diseases, Health Research Institute Marqués de Valdecilla (IDIVAL), 39011 Santander, Spain; marcos.lopez@scsalud.es; 3Biomedical Research Networking Center in Hepatic and Digestive Diseases (CIBERehd), 28029 Madrid, Spain; 4Hematology Department, University Hospital Marqués de Valdecilla, 39008 Santander, Spain; mdelrocio.perez@scsalud.es (M.d.R.P.M.); javier.nunezc@scsalud.es (J.N.C.); mana.batlle@scsalud.es (A.B.L.); 5Radiology Department, University Hospital Marqués de Valdecilla, 39008 Santander, Spain; franciscojose.gonzalezs@scsalud.es; 6Immunology Department, University Hospital Marqués de Valdecilla, 39008 Santander, Spain

**Keywords:** liver cirrhosis, portal vein thrombosis, thrombophilia

## Abstract

In patients with liver cirrhosis the contribution of inherited and acquired prothrombotic disorders in the development of non-malignant portal vein thrombosis (PVT) is inconclusive. The purpose of this retrospective study was to examine the prevalence of thrombophilia in this setting at our center from January 2012 to November 2019. Tests included gene mutational analysis for Factor V Leiden, prothrombin G20210A, *JAK2* (V617F), Calreticulin (*CARL*), in addition to activated protein C resistance, antithrombin III, protein C and S levels, and antiphospholipid antibodies. We included 77 patients, six of whom (7.8%) had a thrombophilic disorder: antiphospholipid syndrome in four patients, prothrombin gene mutation in one and factor V Leiden mutation in one. This latter patient had also been diagnosed with polycythemia vera years before PVT development. Complete thrombosis of the main portal vein and re-thrombosis after stopping anticoagulation were more frequent in patients with thrombophilia, but the rates of recanalization under anticoagulant therapy were similar among groups. No other difference was accounted between groups. The low prevalence of acquired and inherited thrombophilia found in patients with cirrhosis and PVT support testing for these disorders on an individual basis and avoiding universal screening to reduce costs and unwarranted testing.

## 1. Introduction

Non-malignant portal vein thrombosis (PVT) is defined as a thrombus that develops within the portal vein trunk and intrahepatic portal branches, which may also involve the splenic (SV) or superior mesenteric veins (SMV). In the absence of recanalization, the portal venous lumen is obliterated and portoportal collaterals develop resulting in portal cavernoma. Although the latter transformation can develop very early after acute PVT, it is generally used to define the chronic stage of PVT [1]. It constitutes the most common thrombotic event in patients with cirrhosis, with increased rates in the setting of advanced liver disease. The reported prevalence of PVT varies with different diagnostic methods and target populations, ranging between approximately 10–25% in patients with decompensated cirrhosis and 1–5% in those with compensated cirrhosis [2]. Despite being a well-known complication of liver cirrhosis, the contribution of PVT to hepatic decompensation and overall mortality is still a matter of debate [1,3,4,5]. Discrepancies among studies regarding patient selection criteria (compensated vs. decompensated), degree and extent of thrombosis (occlusive vs. nonocclusive), treatment strategies (anticoagulation vs. no anticoagulation), sample size and time of follow-up have led to conflicting data [6]. There is consequently no consensus on its optimal management and no definitive recommendations have been reported in clinical guidelines or consensus conferences [1,4,5,7,8].

The mechanisms involved in the development of PVT in patients with cirrhosis are also not yet fully understood. Of the three pathophysiologic factors predisposing to thrombosis described in the triad of Virchow (i.e., slow blood flow, endothelial damage and hypercoagulability), portal flow seems to be the most influential in the setting of cirrhosis [4]. The efficacy of transjugular intrahepatic portosystemic shunt (TIPS) in restoring PVT patency by presumably increasing portal flow [1] and the identification of a reduced portal flow as a major risk factor for PVT development support this notion [9,10]. Other potential mechanisms involved include a state of hypercoagulability in more advanced disease, bacterial translocation and inflammation, and vascular injury to the portal venous system secondary to several procedures (e.g., splenectomy) [3].

Inherited and acquired prothrombotic disorders may also play a role, although current data are conflicting. The limited number of studies available are mostly case-control studies with small sample sizes. Their study design, target population (diverse ethnicities and geographical locations), diagnostic criteria for PVT, and assessment of thrombophilic conditions vary widely, and contribute to the inconsistent results [11,12,13,14,15,16,17,18,19,20,21,22,23,24,25,26,27,28,29,30,31,32,33]. Moreover, none of these studies have properly evaluated whether the presence of thrombophilia impact the progression rate or response to treatment. Among the different thrombophilic genetic defects, Factor V Leiden (FVL) and prothrombin G20210A (PTHR) mutations have been the most frequently studied. Three meta-analysis concluded that they increased the risk of PVT in patients with cirrhosis [34,35,36], although in one of them this association was not shown for PTHR [35] and all of them are biased by the quality of the studies included. Inherited protein C, protein S or antithrombin III deficiencies are difficult to detect due to co-existent liver synthetic dysfunction [4]. Their levels, however, do not seem to be associated with PVT development [37]. The methylene tetrahydrofolate reductase C677T and plasminogen activator inhibitor– type 1 4G-4G mutations have also been described as independent predictors of PVT [18], although these polymorphisms have not been conclusively associated with increased thrombotic risk [38]. The role of acquired prothrombotic disorders has been less evaluated in patients with liver cirrhosis and PVT. In contrast to non-cirrhotic PVT, the relevance of myeloproliferative disorders and antiphospholipid syndrome (APS) is so far inconclusive [3]. Due to the conflicting data, current guidelines make no strong recommendation regarding testing for these conditions in either a screening capacity before PVT diagnosis, or confirmatory once thrombosis has developed [1,4,7,8].

The main purpose of this study was to examine the prevalence of inherited and acquired thrombophilia in cirrhotic non-malignant PVT at our center and to describe the clinical presentation of PVT in these patients. As a secondary aim we analyzed the course of PVT in the whole cohort and determined the factors associated with PVT recanalization.

## 2. Experimental Section

### 2.1. Patients

The Marques de Valdecilla University Hospital (Santander, Cantabria, Spain) is an urban, academic tertiary care center. Since 2012 we began to test for thrombophilia in patients with cirrhosis who developed non-malignant PVT. In this report, we conducted a retrospective cohort study regarding the presence and influence on outcomes of inherited and acquired thrombophilia in this setting from January 2012 to November 2019. Patients were mainly selected from the database of our Gastrointestinal and Hepatology Service. To ensure the identification of all eligible patients we also reviewed: (1) all thrombophilic studies performed during the study period by the Department of Hematology; (2) hospital discharge records. This search did not include cases of PVT diagnosed at autopsy.

Cirrhosis was confirmed on the basis of clinical, laboratory, and imaging studies or liver biopsy, and PVT was diagnosed as part of biannual screening for hepatocellular carcinoma (HCC) or during hospitalization for decompensated cirrhosis. Patients without thrombophilic study or with malignant PVT (i.e., presence of vascularization of the thrombus at contrast imaging, mass-forming features of PVT and/or evidence of disruption of vessel walls) were excluded. The presence of a cavernomatous transformation of the portal vein was not considered an exclusion criterion.

### 2.2. Definitions

PVT was defined as the absence of flow in part of or in the entire lumen of any site among portal vein trunk, portal vein branches, superior mesenteric vein (SMV) or splenic vein (SV) caused by the presence of solid material within the vein, as documented by an imaging technique (Doppler ultrasound (US), computed tomography (CT]), or magnetic resonance imaging (MRI)).

#### 2.2.1. Thrombosis

Thrombosis was considered complete when the blood flow was absent, or the thrombus involved more than 90% of the vessel diameter. Otherwise, it was defined as partial. Evolution of thrombosis was classified as previously reported by Delgado et al. [39].

#### 2.2.2. Recanalization

Recanalization was considered complete when the portal vein trunk, portal vein branches, SMV, and SV were all completely patent. Recanalization was considered partial when some parts of the thrombus persisted but there was at least a 50% reduction in the thickness or length of the thrombus, or when complete patency was achieved in the portal vein trunk and in at least one of the following segments if previously thrombosed: main intrahepatic branches, SV or SMV. Lack of recanalization according to the definition above was considered to be a non-response to treatment.

#### 2.2.3. Thrombosis Progression

Thrombosis progression was considered to occur when thrombus thickness increased >50% or when the thrombosis extended to previously unaffected segments of the spleno-porto-mesenteric axis.

### 2.3. Anticoagulation Therapy

Treatment decisions were at the discretion of the physician taking care of the patient. Appropriate primary or secondary prophylaxis of variceal bleeding was always undertaken before starting anticoagulant treatment. In general, full-dose low-molecular-weight heparin (LMWH) was started and switched after four–six weeks to vitamin k antagonists to maintain INR between 2 and 3. Patients with significantly prolonged INR in the setting of advanced liver cirrhosis were maintained with LMWH. Among patients receiving LMWH, anti-factor Xa activity (HemosIL liquid Anti Xa) was not routinely performed to verify the efficacy of anticoagulation.

### 2.4. Follow Up

The date of the first abdominal imaging study detecting PVT was considered as time zero for computing follow-up. Clinical, epidemiological, laboratory, and radiological data were collected at PVT diagnosis. Imaging follow-up was not performed according to a strict protocol, but at the discretion of the attending physicians. In general, it consisted of abdominal US and CT/MRI within 6 months of start of anticoagulation and then abdominal Doppler US every 6 months. The most recent follow-up imaging studies were used to evaluate PVT recanalization if performed at least four weeks after PVT diagnosis. Recurrent thrombosis after recanalization was assessed in those patients that stopped anticoagulation therapy before the end study. All data was extracted from the electronic medical record.

### 2.5. Thrombophilic Study

Test for thrombophilia were delayed until at least four weeks after PVT diagnosis, at which time LMWH were switched to vitamin K antagonists. Tests included gene mutational analysis for FVL and PTHR, in addition to activated protein C resistance, antithrombin III, protein C and S levels, and antiphospholipid antibodies. The latter included anticardiolipin (aCL), antibeta2 glycoprotein (aB2GPI), and lupus anticoagulant (LA). Screening of myeloproliferative neoplasms by gene mutational analysis for *JAK2* (V617F) and *CALR* was performed since 2015. The hypercoagulable panel was interpreted by the Hematology department, and the presence of liver cirrhosis was taken into consideration in all patients. Diagnosis of APS and myeloproliferative neoplasms were defined according to the revised Sapporo criteria [40] and the revised WHO classification of myeloid neoplasms [41], respectively.

Blood samples were collected in vacutainer tubes containing NaCitrate 3.2% in 1/9 proportion. After centrifugation (2500 rpm), 1 mL aliquots were stored at −30 °C and used within 30 days. Protein C and antithrombin III were determined using automated chromogenic assay for quantitative determination on IL Coagulation Systems (HemosIL Werfen^®^, Instrumentation Laboratory, Bedford, MA, USA). Free Protein S level was determined using an automated latex ligand immunoassay on IL Coagulation Systems (HemosIL Werfen^®^). Activated Protein C resistance was determined with coagulometric test based on TTPa parameter (HemosIL Werfen^®^). Normal values were established according to 100 control patients of the same age range and gender and were as follows: antithrombin, 85–140%; protein C, 85–140%; protein S, 70–120%. LA was determined using diluted Russell’s viper venom test and silica Cotting time (HemosIL Werfen^®^). Serum IgG and IgM aCL and aB2GPI levels were measured by ELISA following manufacturer’s instructions (Orgentec Diagnostika, Mainz, Germany) and expressed in IgG phospholipid (GPL) or IgM phospholipid (MPL) units or U/mL, respectively. Titers were considered to be positive when they were above the 99th percentile, thus corresponding to values above 20 GPL, MPL or U/mL (medium: 20–30 or high: >30 titers). If positive, they were repeated at least 12 weeks later in order to confirm their positivity.

Prothrombin G20210A and FVQ506 mutation were determined using LightCycler^®^ 2.0 instrument utilizing polymerase chain reaction (PCR Roche Diagnostics^®^, Roche Diagnostics GmbH, Mannheim, Germany) RNA was extracted from peripheral blood leukocytes and was reverse transcribed with random hexamer primers following standard protocol. Allele-specific standard PCR was performed in a final volume of 50 µL using forward primers that were specific for *JAK2* V617F Forward 5′-TCCTCAGAACGTTGATGGCAG-3′ and Reverse 3′-GTTTTACTTACTCTCGTCTCCACAAAA-3′ producing a 279 bp product in positive cases. A positive control was added in each reaction. A control PCR with a Wild type Forward primer was run in parallel to assess the quality of each sample. PCR conditions were 94 °C for 7 min; 40 cycles with denaturation at 94 °C for 30 s, annealing at 60 °C for 30 s, and elongation at 72 °C for 30 s; 1 cycle at 72 °C for 7 min; and a final hold at 4 °C. Verification of the expected PCR product was performed on a 2% agarose gel stained with ethidium bromide. All patient specimens that were negative for *JAK2* V617F were assessed for *CALR* mutations on exon 9 using a polymerase chain reaction (PCR)-based assay (forward primer 5′-GGCAAGGCCCTGAGGTGT-3′ and reverse 5′-GGCCTCAGTCCAGCCCTG-3′ PCR; annealing temperature 55 °C) producing a 263 bp band on a 2% agarose gel. Type 1 mutations (c.1092_1143del) were detected on the gel. To detect *CALR* additional mutations, subsequent sanger deep-sequencing was performed.

The study protocol conformed to the ethical guidelines of the 1975 Declaration of Helsinki as reflected in a priori approval by the Ethics Committee for Clinical Research of Cantabria (internal code: 2020.246). A waiver of informed consent was provided since the study was considered a retrospective review.

### 2.6. Statistical Analysis

Continuous variables were assessed by Kolmogorov-Smirnov test for normality and expressed as mean ± standard deviation or median and interquartile range. Categorical variables were expressed as counts and percentages. Comparisons between patients with and without thrombophilia were performed using Student’s T test or Mann-Whitney U test for continuous variables and χ2 or Fisher’s exact test for categorical variables as applicable. Follow-up was calculated from the time of PVT diagnosis to the date of the last imaging study available before death, liver transplantation, or 1st June 2020. We performed a Cox univariate analysis to explore the variables associated with PVT recanalization (partial or complete). Those variables associated (*p* ≤ 0.10) with PVT recanalization in the univariate analysis or those considered to be clinically significant were tested in a Cox multivariate regression model. We estimated the contribution of each variable by the hazard ratio (HR) with its 95% confidence interval. *P* < 0.05 was considered statistically significant. We limited the number of variables in the multivariable analysis to 1 per 5–10 outcomes. Statistical analysis was performed with IBM SPSS Statistics v22.0 for Mac (IBM Corp., Armonk, NY, USA).

## 3. Results

### 3.1. Prevalence of Thrombophilia

During the study period, 166 cases of PVT were evaluated, of which 89 were excluded (Figure 1). The final cohort included 77 patients with liver cirrhosis and non-malignant PVT in whom a thrombophilic study had been performed. Compared to patients excluded for not having a thrombophilia workup, the patients included were younger (61.9 (55.0–67.6) vs. 70.0 (58.2–77.1), *p* = 0.003), had lower Child (7 (6–9) vs. 13 (10–19.5); *p* = 0.014) and MELD points (12 (10–14) vs. 13 (10–19.5)) and had a lower prevalence of HCC at PVT diagnosis (13% vs. 32%; *p* = 0.014). Screening of *JAK2* V617 and *CARL* mutations was investigated in 37 patients (48.1%) and antiphospholipid antibodies were not tested in one patient. The remaining thrombophilic tests were available in the whole cohort.

Six patients (7.8%) had a thrombophilic disorder: antiphospholipid syndrome in four patients, PTHR mutation in one and FVL mutation in one. This latter patient had already been diagnosed with polycythemia vera (*JAK2* V617F positive) years before PVT development. A detailed description of these six patients is provided in Table 1.

Antiphospholipid antibodies were present in 13 patients (16.9%), but eventually only four were diagnosed with APS once their persistent positivity was confirmed in further tests after 12 weeks. LA and IgM aB2GPI were the most frequent antiphospholipid antibodies (6.6% and 7.9%, respectively). Their positivity tended to be more frequent in Child B-C patients in comparison to Child A patients (22.9 vs. 7.4%, *p* = 0.118). Levels of antithrombin III, protein S and C were decreased in 70 (90.9%), 72 (93.5%) and 44 (57.1%) patients, respectively. These deficiencies were interpreted as secondary to liver cirrhosis and not as an inherited thrombophilia.

### 3.2. Characteristics of Patients with and without Thrombophilia

The clinical and epidemiological profile of patients with and without thrombophilia was similar (Table 2). Most patients had an alcoholic liver disease, were decompensated before PVT diagnosis and were on non-selective betablocker treatment in the setting of primary or secondary prophylaxis of variceal bleeding. Three patients, all in the non-thrombophilic group, had suffered a previous arterial or venous thrombotic event (myocardial infarction in one patient, deep vein thrombosis in another, and pulmonary embolism in the remaining patient). No female patient in either group had a history of pregnancy complications consistent with thrombophilia.

### 3.3. Extension and Clinical Characteristics of Thrombosis at Diagnosis

Diagnosis of PVT was made by CT or MRI in the majority of patients. The portal vein or its branches were the only thrombosed vessels in 50 patients (64.9%). In two patients (2.6%) the thrombosis extended to the SV, in 17 (22.1%) to the SMV, and in four patients (5.2%) it involved the entire splenoportomesenteric venous axis. Three patients (3.9%) had isolated thrombosis of the SMV and one (1.3%) of the SV. Portal cavernoma was established in nine patients (11.7%). In most cases, thrombosis was partial, regardless of its location. In patients with thrombophilia, however, complete thrombosis of the main portal vein was more frequent in comparison to patients without thrombophilia (Table 3). Four patients, all in the non-thrombophilic group, had a local predisposing local factor. In all them PVT developed several weeks after radiofrequency ablation of HCC.

Thirty-three patients (42.9%) showed new symptoms overlapping with the diagnosis of PVT. The most frequent decompensation event was variceal bleeding followed by hepatic encephalopathy. PVT only led to the development of mesenteric ischemia in one patient (1.5%). No differences in clinical presentation or analytical parameters were observed between patients with and without thrombophilia (Table 3).

### 3.4. Treatment and Factors Associated with the Outcome of Portal Vein Thrombosis

Five patients from the non-thrombophilic group were excluded from this analysis. Four of them were participating in a randomized control trial to evaluate the effect of rivaroxaban in patients with advanced liver disease with PVT (Tromboxaban; EudraCT Number 2016-003240-37) and the other patient died soon after PVT diagnosis due to septic shock. In the remaining 72 patients, anticoagulation was frequently started (80.5%), while TIPS placement was rare (5.6%) and always indicated by complications of cirrhosis, and not by PVT progression (Table 4).

Therapy was maintained with LMWH in 12 patients, with vitamin K antagonists in 45, and with apixaban in one patient. Median delay from PVT diagnosis to the beginning of anticoagulation treatment was nine days (interquartile range, 0–42 days) and its median duration was 12.6 months (interquartile range, 6.2–27.0). PVT recanalization was similar between patients with and without thrombophilia both in treated and non-treated patients (Table 4).

Duration of anticoagulation was finite in thirty-one patients (53.4%). After ceasing anticoagulation, re-thrombosis developed in ten patients (32.3%), with a trend for this event to occur more frequently in patients with thrombophilia (100% vs. 27.6%; *p* = 0.097). No other difference regarding treatment and outcome was observed between patients with and without thrombophilia (Table 4).

Regarding variables associated with PVT recanalization, the only variable that remained significant on multivariate analysis was the presence of cavernomatosis, which was associated with a worse PVT outcome (HR (95% CI): 0.110 (0.014–0.849), *p* = 0.034).

## 4. Discussion

In patients with liver cirrhosis, the contribution of inherited and acquired prothrombotic disorders in the development of non-malignant PVT is inconclusive. The limited available data is hampered by the heterogeneity and small sample size of the studies (Table 5). The present report constitutes one of the largest series on this topic and, contrary to most published studies, includes a thorough thrombophilia workup. The interpretation of the hypercoagulable panel by the Hematology Department is another strength of the present work as many previous studies do not provide information in this regard. Our results show a low prevalence of inherited and acquired thrombophilia in patients with cirrhotic non-malignant PVT and question the utility of universal screening in this setting.

APS was the most common thrombophilic disorder found in our cohort (5.3%). Similar figures have been observed in some [29,33], but not all studies [15,23,24,39]. These discrepancies might be explained not only by differences in study design or target population, but also by a low adherence to the revised APS criteria [40]. Indeed, many studies performed antiphospholipid-antibody testing only once and cutoffs varied among them. Their clinical significance in the setting of cirrhosis, however, is unclear as antiphospholipid-antibody positivity can be frequently found in patients with liver disease without any evidence of venous thrombosis. This has been regarded as an epiphenomenon of chronic liver injury, and in line with this premise a recent meta-analysis did not find a clear association between the presence of antiphospholipid antibodies and development of PVT in patients with liver cirrhosis [42]. These findings together with the frequently reduced levels of antithrombin III, protein S and C encountered in patients with liver disease (in our cohort 90.9%, 93.5% and 57.1%, respectively) highlight the difficult assessment of the hypercoagulable panel in the setting of cirrhosis and the need for this assessment to be performed by the Hematology Department. The presence of inherited thrombophilia was limited to two patients (2.6%). These figures are not higher than those described for these disorders in the general population [43]. Finally, the infrequent presence of myeloproliferative neoplasm (2.7%) is in line with previous studies in this setting [21,22,26,31,33]. Overall, our data show a low prevalence of thrombophilia, either inherited or acquired, which is in agreement with other previous reports that did not find an association between the presence of thrombophilia and the risk of PVT in patients with liver cirrhosis [11,16,21,22,24,27,28,30,33,39].

Regarding the influence of these disorders on the clinical presentation of PVT and response to anticoagulation, complete thrombosis of the main portal vein and re-thrombosis after stopping anticoagulation were more frequent in patients with thrombophilia. In contrast, the rates of recanalization under anticoagulant therapy were similar among groups and in keeping with those reported to date [44]. The low number of patients with thrombophilia limits the significance of these findings and call for larger studies to properly address these issues. While awaiting further evidence, the currently available data along with associated healthcare costs support performing these tests on an individual basis. As in patients with venous thromboembolic disease [43], it is not possible to give a validated recommendation on how such patients should be selected for testing. Our current strategy limits the screening for patients with family histories of prothrombotic defects, patients with multiple sites of thrombosis, recurrent thrombosis, or when treatment decisions (i.e., anticoagulation duration) may be affected.

The main limitations of our study are its retrospective unicenter design and its relatively small sample size, which may limit the external validity of our results and undermine the power to assess the factors associated with PVT recanalization. The exclusion of a significant number of patients without a thrombophilic study who differed in relevant clinical variables from the study population may further introduce a selection bias. We do believe, however, that most of these exclusions are due to a progressive adherence of physicians to conducting a thrombophilic study in this setting. Thus, 76% of these exclusions involve PVT cases that were diagnosed during the first three years of the implementation of the study protocol. The absence of a control group of patients with cirrhosis but without PVT should also be acknowledged. Nonetheless, the overall interpretation of our results would probably not change given the low prevalence of thrombophilic disorders found in our cohort. This low prevalence makes the influence of thrombophilia on the outcome of PVT unreliable to evaluate. Finally, *JAK2* and *CALR* mutations were not performed in all patients as these tests were later included in the thrombophilia workup. Of note, the PCR used in this study did not allow the quantitation of the allelic burden nor was able to detect low burden cases (below around 5%). Investigation of the allelic burden with more sensitive techniques such as digital PCR provides more information in this regard. Moreover, investigation of additional somatic variants in *JAK2*, in other myeloproliferative associated genes such as MPL or in genes involved in the age and clonal hematopoiesis that have been associated with an increased risk of cardiovascular disease, particularly in DNMT3A, TET2, and ASXL1, could have provided more valuable data [45].

## 5. Conclusions

We found a low prevalence of acquired and inherited thrombophilia in patients with cirrhosis and PVT. Our results support testing for these disorders on an individual basis and avoiding universal screening to reduce costs and unwarranted testing. Future prospective studies integrating evaluation of liver disease stage, local and genetic factors are needed to identify individualized criteria to perform these tests and to evaluate their impact on the progression rate or response to treatment.

## Figures and Tables

**Figure 1 jcm-09-02822-f001:**
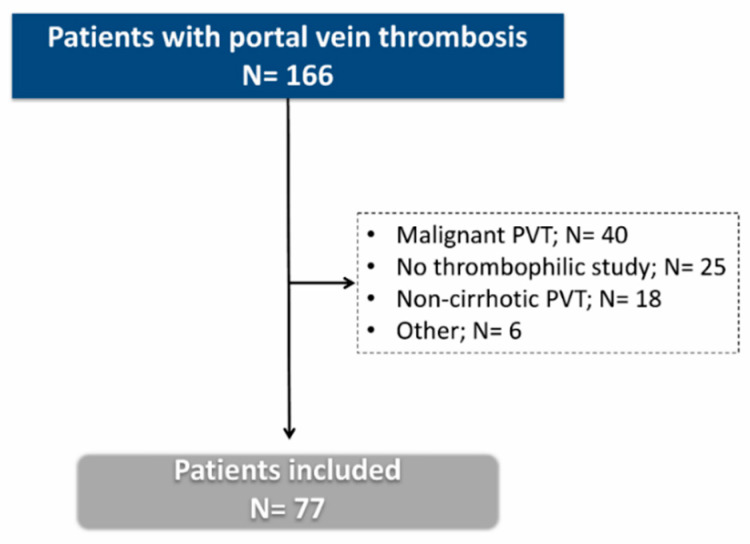
Flowchart of the study selection process. PVT: portal vein thrombosis.

**Table 1 jcm-09-02822-t001:** Clinical, imaging, endoscopic and laboratory features of patients with thrombophilia.

Variable *	Case 1	Case 2	Case 3	Case 4	Case 5	Case 6
Type of thrombophilia	APS	APS	APS	APS	PV + FVL (heterozygous)	PTHR (heterozygous)
Age (years)	51	77	57	56	72	69
Gender	Female	Male	Male	Male	Male	Male
Race	Caucasian	Caucasian	Caucasian	Caucasian	Caucasian	Caucasian
Comorbidity	Diabetes	Diabetes	No	Hypertension	Diabetes	Diabetes
Etiology of liver disease	Hepatitis C	Alcohol	Alcohol	Alcohol	Alcohol	Alcohol
Child-Pugh	B (7 points)	B (7 points)	A (6 points)	B (9 points)	B (7 points)	A (6 points)
MELD (points)	11	9	8	27	13	10
Previous decompensation	VB + HE + ascites	Ascites	VB + ascites	Ascites	No	VB + HE + ascites
EV without bleeding		High risk		Low risk	High risk	
Non-selective betablockers	Yes	Yes	Yes	Yes	No	Yes
Hepatocellular carcinoma (HCC) (No/BCLC stage)	No	No	No	No	No	A
Previous thrombotic events	No	No	No	No	No	No
Imaging for PVT diagnosis	CT	US	CT	CT	US	CT
Localization and extension	Main PV/Complete	Main PV/Partial	Main PV (complete), SV and SMV (partial)	PV, both branches and SMV/Partial	PV and branches/Complete	Right PV/Complete
Portal cavernoma	Yes	No	No	No	No	No
Local predisposing factor	No	No	No	No	No	No
Decompensation at diagnosis	Ascites	VB and ascites	No	SBP and HE	Ascites	No
Other symptoms	No	No	No	No	No	No
Analytical parameters at diagnosis						
Leucocytes (×103 μL)	3.4	5.1	3.0	12.0	2.3	5.0
Platelets (×103 μL)	60	140	35	94	65	162
Hemoglobin (gr/dL)	9.8	11.6	14.8	13.8	11	13.9
Creatinine (mg/dL)	0.96	0.97	0.80	2.81	0.88	0.85
Sodium (mEq/L)	138	134	137	129	138	137
ALT (U/L)	56	44	20	21	62	26
AP (U/L)	119	235	46	100	47	123
Bilirubin (mg/dL)	0.9	1.2	0.6	3	2.1	1.4
Albumin (gr/dL)	2.9	3.2	4.4	3.4	4.2	3.8
INR	1.45	1.14	1.2	1.77	1.34	1.27
Treatment	Acenocoumarol	Acenocoumarol	Acenocoumarol	Acenocoumarol	LMWH	No
PVT evolution	Progression	Partial resolution	No	Total resolution	Total resolution	Stability
Duration anticoagulation (months)	Indefinite (157.2)	Finite (8.1)	Indefinite (101.1)	Indefinite (38.6)	(Finite) 22.2	
Re-thrombosis		Yes			Yes	
Exitus/LT	LT	Death	Death	No	Death	LT
Time of follow-up (months)	21.3	16.9	102.5	38.6	85.0	24.0

* Quantitative variables were expressed as median and interquartile range and qualitative variables as absolute value (proportion).

**Table 2 jcm-09-02822-t002:** Clinical and epidemiological profile of patients in the whole cohort and in patients with and without thrombophilia.

Variable *	Population (N = 77)	Non-Thrombophilia (N = 71)	Thrombophilia (N = 6)	*p*
Age (years)	61.9 (55.0–67.6)	61.9 (54.9–67.2)	63 (54.6–73.6)	0.464
Gender (male)	67 (87)	62 (87.3)	5 (75.0)	0.579
Race (Caucasian)	76 (98.7)	70 (98.6)	6 (100)	1
Diabetes Mellitus	24 (31.2)	20 (28.2)	4 (66.7)	0.072
Dyslipidemia	12 (15.6)	12 (16.9)	0 (0)	0.582
Arterial hypertension	22 (28.6)	21 (29.6)	1 (16.7)	0.668
Chronic kidney injury	5 (6.5)	5 (7.0)	0 (0)	0.929
HIV	4 (5.2)	4 (5.6)	0 (0)	1
Etiology of liver disease				0.969
Alcohol	49 (63.6)	44 (62.0)	5 (83.3)	
Hepatitis C	8 (10.4)	7 (9.9)	1 (16.7)	
Other	20 (26.0)	20 (28.2)	0 (0)	
Child-Pugh (points)	7 (6–9)	7 (6–9)	7 (5.8–7.5)	0.379
Child A/B/C (%)	35/50/15	36/49/15	33/67/0	0.519
MELD (points)	12 (10–14)	13 (10–14)	11 (9.0–15.8)	0.814
Previous TIPS	4 (5.2)	4 (5.6)	0 (0)	1
Liver allograft cirrhosis	3 (3.9)	3 (4.2)	0 (0)	1
Esophageal varices (Low/High risk)	14 (32)/22 (50)	13 (31.7)/20 (48.8)	1 (33.3)/2 (66.7)	0.682
Previous variceal bleeding	33 (42.9)	30 (42.3)	3 (50)	1
Non-selective betablockers	51 (66.2)	46 (64.8)	5 (83.3)	0.657
Previous ascites (No/Yes/Refractory) (%)	27/65/8	28/63/9	17/83/0	0.818
Previous SBP	6 (7.8)	6 (8.5)	0 (0)	1
Previous HE (No/Episodic/Recurrent) (%)	75/24/1	76/23/1	67/33/0	0.808
Any previous decompensation	62 (80.5)	57 (80.3)	5 (83.3)	1
HCC (No/BCLC stage A/B) (%)	87/12/1	88/11/1	83/17/0	0.890
Previous arterial/venous thrombotic events	3 (3.9)	3 (4.2)	0 (0)	1

* Quantitative variables were expressed as median and interquartile range and qualitative variables as absolute value (proportion).

**Table 3 jcm-09-02822-t003:** Extension and clinical characteristic of portal vein thrombosis at diagnosis in the whole cohort and in patients with and without thrombophilia.

Variable *	Population (N = 77)	Non-Thrombophilia (N = 71)	Thrombophilia (N = 6)	*p*
CT or MRI portal vein thrombosis (PVT) diagnosis	67 (87.0)	63 (88.7)	4 (66.7)	0.172
Localization and extension				
Right PV (Partial/total) (%)	27 (35.1)/8 (10.4)	26 (36.6)/6 (8.5)	1 (16.7)/2 (33.3)	0.139
Left PV (Partial/total) (%)	18 (23.4)/5 (6.5)	16 (22.5)/5 (7.0)	2 (33.3)/0 (0)	0.701
Main PV (Partial/total) (%)	49 (63.6)/8 (10.4)	47 (66.2)/5 (7.0)	2 (33.3)/3 (50)	0.004
Splenic vein (SV) (Partial/total) (%)	5 (6.5)/2 (2.6)	4 (5.6)/2 (2.8)	1 (16.7)/0 (0)	0.536
Superior mesenteric vein (SMV) (Partial/total) (%)	21 (27.3)/3 (3.9)	19 (26.8)/3 (4.2)	2 (33.3)/0 (0)	0.841
Portal cavernoma	9 (11.7)	8 (11.3)	1 (16.7)	0.538
Local predisposing factor	4 (5.2)	4 (5.6)	0 (0)	1
Symptoms at diagnosis	33 (42.9)	31 (43.7)	2 (33.3)	0.695
Acute mesenteric ischemia	1 (1.3)	1 (1.4)	0 (0)	1
Abdominal pain	7 (9.1)	7 (9.9)	0 (0)	1
Fever	2 (2.6)	2 (2.8)	0 (0)	1
Variceal bleeding	14 (18.2)	13 (18.3)	1 (16.7)	1
Ascites (total/de novo)	38 (49.4)/5 (6.5)	34 (47.9)/4 (5.6)	4 (66.7)/1 (16.7)	0.431
SBP	6 (7.8)	5 (7.0)	1 (16.7)	0.396
HE	10 (13.0)	9 (12.7)	1 (16.7)	0.579
Analytical parameters at diagnosis				
Leucocytes (×103 μL)	5.0 (3.3–6.0)	5.0 (3.4–6.0)	4.2 (2.8–6.8)	0.791
Platelets (×103 μL)	79 (62–110)	79 (63–109)	80 (54–146)	0.882
Hemoglobin (gr/dL)	12.8 (10.4–14.4)	12.8 (10.3–14.4)	12.7 (10.7–14.1)	0.905
Creatinine (mg/dL)	0.8 (0.7–1.0)	0.8 (0.7–1.0)	0.9 (0.8–1.4)	0.309
Sodium (mEq/L)	139 (137–141)	139 (137–141)	137 (132–138)	0.124
ALT (U/L)	34 (23–46)	34 (24–45)	35 (21–58)	0.768
Alkaline phosphatase (U/L)	114 (78–154)	114 (79–155)	110 (47–151)	0.576
Bilirubin	1.6 (1.1–2.5)	1.6 (1.2–2.6)	1.3 (0.8–2.3)	0.389
Albumin (gr/dL)	3.4 (3.0–3.8)	3.4 (3.0–3.8)	3.6 (3.1–4.3)	0.296
INR	1.34 (1.23–1.52)	1.34 (1.23–1.52)	1.31 (1.19–1.53)	0.691

* Quantitative variables were expressed as median and interquartile range and qualitative variables as absolute value (proportion).

**Table 4 jcm-09-02822-t004:** Treatment and outcome of portal vein thrombosis in the whole cohort and in patients with and without thrombophilia.

Variable *	Population (N = 72)	Non-Thrombophilia (N = 66)	Thrombophilia (N = 6)	*p*
Anticoagulation	58 (80.6)	53 (80.3)	5 (83.3)	1
Acenocoumarol	45 (77.6)	41 (77.4)	4 (80.0)	0.951
LMWH	12 (20.7)	11 (20.8)	1 (20.0)	
Apixaban	1 (1.7)	1 (1.9)	0 (0)	
Duration (months)	12.6 (6.2–27.0)	11.6 (5.8–20.3)	38.6 (15.1–129.1)	0.174
Transjugular intrahepatic portosystemic shunt (TIPS)	4 (5.6)	4 (6.1)	0 (0)	1
PVT evolution in non-treated patients	10 (3.9)	9 (13.6)	1 (16.7)	0.923
Stability	7 (70.0)	6 (66.7)	1 (100)	
Progression	2 (20.0)	2 (22.2)	0 (0)	
Partial resolution	1 (0.0)	1 (11.1)	0 (0)	
Total resolution	0 (0)	0 (0)	0 (0)	
PVT evolution in treated patients (anticoagulation or TIPS)	62 (86.1)	57 (86.4)	5 (83.3)	0.954
Stability	14 (22.6)	13 (22.8)	1 (20.0)	
Progression	7 (11.3)	6 (10.5)	1 (20.0)	
Partial resolution	12 (19.4)	11 (19.3)	1 (20.0)	
Total resolution	29 (46.8)	27 (47.4)	2 (40.0)	
Re-thrombosis after ceasing anticoagulation	10 (32.3)	8 (27.6)	2 (100)	0.097
Exitus	36 (50.0)	33 (50)	3 (50)	1
Liver transplantation	17 (23.6)	15 (22.7)	2 (33.3)	0.621
Time of follow-up (months)	27.0 (10.9–55.5)	27.0 (10.8–55.0)	22.7 (18.0–69.7)	0.339

* Quantitative variables were expressed as median and interquartile range and qualitative variables as absolute value (proportion).

**Table 5 jcm-09-02822-t005:** Large studies evaluating the prevalence of acquired and inherited thrombophilia in non-malignant portal vein thrombosis in patients with liver cirrhosis *.

Author and Year	N ^†^	Study Period And Type	Population	PTHR ^‡^	FVL ^‡^	APS ^‡^	JAK2 ^‡^	MTHFR ^‡^	PAI ^‡^	Comments
Mahmoud et al.; 1997 [11]	32	NS Retrospective	UK		1/32 (3.1%)					Authors concluded Factor V Leiden (FVL) was not a major contributor of portal vein thrombosis (PVT). Not all 32 patients had liver cirrhosis.
Amitrano et al.; 2000 [12]	23	1998–1999 Case-control	Italy	8/23 (34.8%)	3/13 (13%)	NS		10/23 (43.5%)		Prothrombin G20210A (PTHR) and MTHFR were strongly associated with PVT. ACA in 4% and LA in 0%. No further test to confirm ACA positivity.
Amitrano et al.; 2004 [13]	79	1998–2002 Case-control	Italy	15/70 (21.4%)	8/70 (11.4%)	NS		15/70 (21.4%)		ACA IgG and ACA IgM at low levels in PVT and in one above 40 UI/L. PTHR increased more than fivefold the risk of PVT.
Mangia et al.; 2005 [16]	43	1997–1999 Case-control	Italy	2/43 (4.7%)	1/43 (2.3%)			9/43 (20.9%)		PTHR, FVL and MTHFR were evenly distributed among patients with and without PVT.
Amitrano et al.; 2006 [14]	78	1998–2002 Case-control	Italy	17/78 (21.4%)						PTHR was associated with PVT, and factor II levels were higher in patients with PTHR and PVT.
Pasta et al.; 2006 [17]	78	2000–2005 Case-control	Italy					19/78 (24.4%)		MTHFR was associated with PVT development.
Colaizzo et al.; 2008 [19]	91	NS Retrospective	Italy				5/91 (5.5%)			Authors suggested to search for *JAK2* in the setting of severe PVT, previous thrombosis and no thrombopenia.
Gabr et al.; 2010 [20]	21	NS Case-control	Egypt					7/21 (33%)		Authors concluded that MTHFR was associated with an increased risk of PVT.
Amitrano et al.; 2011 [15]	50	NS Case-control	Italy			0/50 (0%)				Antiphospholipid antibodies played no role in PVT associated with liver cirrhosis.
Ayala et al.; 2012 [21]	50	2001–2006 Case-control	Spain	1/49 (2%)	1/49 (2%)		0/50 (0%)	7/48 (14.6%)		No association was observed between pre-transplant PVT and presence of genetic thrombophilia.
Delgado et al.; 2012 [39]	43	2003–2010 Retrospective	Spain	3/43 (7%)	1/43 (2.3%)	1/43 (2.3%)				Multicenter study. Thrombophilia in 16% of patients and it was not associated with response to anticoagulation.
Qi et al.; 2012 [22]	71	2009–2011 Prospective	China				1/71 (1.4%)			Prevalence very close to that of a Chinese hospital population of patients without PVT.
Senzolo et al.; 2012 [23]	56	2007–2008 Prospective	UK, Italy	4/56 (7%)	2/56 (3.6%)	0/56 (0%)	-	-	-	Bicenter study. One patient had combined thrombophilia (FVL + PTHR).
Werner et al.; 2013 [24]	69	2005–2011 Retrospective	USA	0/22 (0%)	0/22 (0%)	0/22 (0%)				One patient had antithrombin deficiency.
Karakose et al.; 2015 [26]	38	2005–2009 Prospective	Turkey	4/38 (10.5%)	5/38 (13.1%)		1/38 (2.6%)	5/38 (13.2%)		Unicenter study.
Nery et al.; 2015 [25]	67	2000–2006 RCT	France	NS	NS					Multicenter RCT. PTHR and FVL were studied in 283 patients, (PVT in 67). Their presence was not associated with PVT.
Saugel et al.; 2015 [27]	21	2009–2011 Case-control	Germany	0/21 (0%)	1/21 (4.8%)		2/21 (9.5%)			There was a trend for higher frequency of *JAK2* mutation in cirrhotic patients with PVT than those without PVT.
Lancelloti et al.; 2016 [28]	24	2013 Case-control	Italy	1/24 (4.2%)	0/24 (0%)	NS				PTHR and FVL were infrequent and not associated with PVT development.
Pasta et al.; 2016 [18]	350	2000–2014 Prospective	Italy	18/350 (5%)	29/350 (8%)			88/350 (25%)	111/350 (31%)	Data from 3 prospective studies. ≥1 genetic thrombophilia in 54% of patients. MTHFR/PAI were associated with PVT.
Ventura et al.; 2016 [29]	38	2009–2013 Case-control	Italy	11/38 (10.5%)	4/38 (10.5%)	2/38 (7.9%)		13/38 (34.2)		PTHR and hyperhomocysteinemia were associated with PVT development.
Artaza et al.; 2018 [30]	32	2009–2015 Retrospective	Spain	0/24 (0%)	2/24 (8.3%)		1/24 (4.2%)			Thrombophilia in 4 patients (16%). No association between thrombophilia and evolution of PVT.
Senzolo et al.; 2018 [31]	149	2008–2012 Prospective	International	7/64 (10.9%)	7/71 (9.9%)		1/32 (3.1%)			Thrombophilia testing <50% of the patients. Authors did not search for an association between PVT and thrombophilia.
Cagin et al.; 2019 [32]	98	2009–2015 Case-control	Turkey	15/98 (15.3%)	12/98 (12.2%)			16/98 (16.3%)		FVL mutation was the only type of thrombophilia associated with PVT.
Tremblay et al., 2020 [33]	73	2000–2019 Retrospective	USA	4/63 (6.3%)	4/65 (6.1%)	2/66 (3%)	1/45 (2.2%)	1/27 (3.7%)	20/34 (58.8%)	Thrombophilia testing was not complete in most patients and infrequently led to change in management.

* The minimum number of patients with portal vein thrombosis to consider a study as large was 20. ^†^ Denotes the number of patients that developed portal vein thrombosis within each study, not to the total cohort in each of them. ^‡^ Percentages are calculated based on the number of patients tested for each type of thrombophilia, not the total cohort.

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
