# Peer review of "Portal Thrombosis in Cirrhosis: Role of Thrombophilic Disorders"

_jcm, 2020, doi:10.3390/jcm9092822_

Round 1

Reviewer 1 Report

This is a very thorough and well-written retrospective study. The information is helpful for clinicians treating PVT in cirrhosis. I have just a few questions.

‘……during the study period by the Department of Hematology; 3) Hospital discharge records.’……Did this include autopsy results? Apparently you excluded one death from the nonthrombotic group. Why? Were any deaths from the thrombosis group excluded or not reported?

‘No female patient in either group had history of abortion.’ Should you clarify this sentence? Maybe something like: No female patient in either group had history of pregnancy complications consistent with thrombophilia.

Reviewer 2 Report

Portal thrombosis in cirrhosis: role of thrombophilic disorders

Summary:

  • The authors evaluated 77 cirrhotic patients diagnosed with non-malignant PVT during a period of 7 years in a single institution to assess the prevalence of thrombophilic disorders and to explore differences in the clinical presentation and response to AC in patients with and without thrombophilia.
  • While this is a relevant topic and the authors present interesting information; there are several discrepancies in the description of the aims and the methods/statistical analysis performed that should be revisited.   

Comments:

  • The study design should be stated in the methods section. The main objective suggests that the authors had planned a cross-sectional study to evaluate prevalence and to describe clinical characteristics in patients with and without thrombophilia. However, several parts in the text suggest that the authors were interested in assessing the effect of exposure to AC in a retrospective cohort. For instance:
    • In results, a comparison is presented between AC and non-AC groups in cirrhotics with non-malignant PVT. This introduces selection bias because the characteristics of this two groups are not presented (ie: there might be an unequal distribution of thrombophilics in AC vs non-AC or any other potential confounders). The aim of the trial was to compare pts with thrombophilia vs no-thrombophilia (not AC vs non-AC)
    • In statistical design, they mention a logistic regression for the main endpoint of “partial/complete recanalization”. Recanalization was not stated as the main outcome of this study.

  • The statistical analysis does not match the study aims. It is stated that “The correlation between levels of natural anticoagulants and prognostic scores of cirrhosis was estimated by the Pearson correlation coefficient” but this is not an aim of the study.

  • In figure 1, it would be important to state what was the underlying etiology of patients excluded for not having a thrombophilic study. If these were patients with cirrhosis and non-malignant PVT: Do these patients differ in clinical characteristics compared to the study population? If so, this could be another source for selection bias.

  • In results, I would not affirm that there is a correlation, as r=0.25 and -0.36.

  • Would suggest presenting the baseline characteristics of the study population in Table 1, instead of a table with literature review (can be added at the end).

  • Limitations should include selection bias and the fact that the low "n" in the thrombophilic group made the "exploratory secondary aims" unfeasible to evaluate. 

Round 2

Reviewer 2 Report

The authors addressed all the suggestions/comments appropriately.